# Bendiocarb and Malathion Resistance in Two Major Malaria Vector Populations in Cameroon Is Associated with High Frequency of the G119S Mutation (Ace-1) and Overexpression of Detoxification Genes

**DOI:** 10.3390/pathogens11080824

**Published:** 2022-07-23

**Authors:** Idriss Nasser Ngangue-Siewe, Paulette Ndjeunia-Mbiakop, Nelly Armanda Kala-Chouakeu, Roland Bamou, Abdou Talipouo, Landre Djamouko-Djonkam, John Vontas, Konstantinos Mavridis, Jeannette Tombi, Timoléon Tchuinkam, Jean Arthur Mbida-Mbida, Christophe Antonio-Nkondjio

**Affiliations:** 1Laboratory of Animal Biology and Physiology, University of Douala, Douala P.O. Box 24157, Cameroon; idrissngangue@gmail.com (I.N.N.-S.); mbidajean@yahoo.fr (J.A.M.-M.); 2Laboratoire de Recherche sur le Paludisme, Organisation de Coordination pour la lutte contre les Endémies en Afrique Centrale (OCEAC), Yaoundé P.O. Box 288, Cameroon; paulette.ndjeunia@facsciences-uy1.com (P.N.-M.); nelly_kala25@yahoo.com (N.A.K.-C.); bamou2011@gmail.com (R.B.); atalipouo@gmail.com (A.T.); djamoukolandry@yahoo.fr (L.D.-D.); 3Faculty of Sciences, University of Yaoundé I, Yaoundé P.O. Box 812, Cameroon; jeannette.tombi@facsciences-uy1.com; 4Vector-Borne Diseases Laboratory of the Applied Biology and Ecology Research Unit (VBID-URBEA), Department of Animal Biology, Faculty of Science of the University of Dschang, Dschang P.O. Box 067, Cameroon; timotchuinkam@yahoo.fr; 5Institute of Molecular Biology and Biotechnology, Foundation for Research and Technology-Hellas, 70013 Heraklion, Greece; vontas@imbb.forth.gr (J.V.); mavridiskos@gmail.com (K.M.); 6Pesticide Science Laboratory, Department of Crop Science, Agricultural University of Athens, 11855 Athens, Greece; 7Vector Biology, Liverpool School of Tropical Medicine, Pembroke Place, Liverpool L3 5QA, UK

**Keywords:** *An. gambiae*, *An. coluzzii*, carbamates, organophosphate, resistance, ace-1 (G119S) mutation, overexpression, Cameroon

## Abstract

The spread of pyrethroid resistance in malaria vectors is a major threat affecting the performance of current control measures. However, there is still not enough information on the resistance profile of mosquitoes to carbamates and organophosphates which could be used as alternatives. The present study assessed the resistance profile of *Anopheles gambiae* s.l. to bendiocarb and malathion, at the phenotypic and molecular levels, in different eco-epidemiological settings in Cameroon. *Anopheles gambiae* s.l. mosquitoes were collected from four eco-epidemiological settings across the country and their susceptibility level to bendiocarb and malathion was determined using WHO tubes bioassays. The ace-1 target site G119S mutation was screened by PCR. Reverse Transcription quantitative PCR 3-plex TaqMan assays were used to quantify the level of expression of eight genes associated with metabolic resistance. Resistance to malathion and/or bendiocarb was recorded in all study sites except in mosquitoes collected in Kaélé and Njombé. The Ace-1 (G119S) mutation was detected in high frequencies (>40%) in Kékem and Santchou. Both *An. gambiae* and *An. coluzzii* were detected carrying this mutation. The cytochrome P450s gene Cyp6p3 associated with carbamate resistance and the glutathione S-transferase gene Gste2 associated with organophosphate resistance were found to be overexpressed. Genes associated with pyrethroid (*Cyp6m2*, *Cyp9k1*, *Cyp6p3*) and organochlorine (*Gste2*, *Cyp6z1*, *Cyp6m2*) and cuticle resistance (*Cyp4g16*) were also overexpressed. The rapid spread of resistance to organophosphates and carbamates could seriously compromise future control strategies based on IRS. It is therefore becoming important to assess the magnitude of bendiocarb and malathion resistance countrywide.

## 1. Background

Since their discovery, synthetic insecticides have been largely used for the prevention of vector-borne diseases, due to their high efficacy and ease of use [1]. The fight against malaria vectors in Africa mainly relies on the use of insecticide-based interventions, such as indoor residual spraying (IRS) and insecticide-treated mosquito nets (ITNs). 

Pyrethroids, due to their high efficacy against insects and low toxicity to mammals, including humans, are the only insecticide recommended for bed net impregnation [2]. Over 3 billion impregnated bed nets have been distributed across the world since 2010 [3]. However, overreliance on pyrethroids in both public health and agriculture has exerted intensive selective pressure on mosquitoes and has led to the emergence and expansion of pyrethroid resistance in mosquito populations [4,5,6,7]. The increasing resistance of vector species to pyrethroids is a major challenge for malaria control [8]. In several countries in Africa, kdr target site mutations, associated with pyrethroid resistance, are close to fixation and cytochrome P450 genes, encoding for pyrethroid metabolizers, are overexpressed in mosquito populations [9]. The rapid spread of pyrethroid resistance could reverse progress achieved so far in malaria control [10,11].

Carbamates and organophosphates are considered as possible alternatives for controlling pyrethroid-resistant mosquitoes [12,13]. In order to manage insecticide resistance and maintain the effectiveness of current control tools, the World Health Organization (WHO) recommends the implementation of insecticide resistance management strategies, including rotation strategies with the use of two or more insecticides with different mode of action at different periods or the combination of different control interventions [14]. Since 2006, several African countries have started combining long-lasting insecticidal nets (LLINs) and indoor residual spray (IRS) with carbamate or organophosphate for effective control of malaria vector populations [8,15]. Although promising results had been initially recorded, with improved control of vectors and malaria transmission [8], there is a growing concern since carbamate and to some extent organophosphate resistance has begun to be widely distributed in West Africa [16,17,18] and is currently also spreading in Central Africa [19,20]. LLINs are the main tools used for malaria control in Cameroon, but the country is considering complementing the use of LLINs by the addition of indoor residual spraying with organophosphate in places experiencing a high malaria burden or high seasonal transmission of the disease [8]. Carbamates and organophosphates are largely used in agriculture but have never been used in public health in Cameroon and there is, so far, not enough data on the resistance profile of mosquitoes to these insecticides. Resistance to carbamates and organophosphates is associated with the presence of the G119S mutation and the overexpression of detoxification genes [18,19,21,22]. Alarmingly, recent studies in Cameroon have indicated the emergence of resistance to carbamate and organophosphates, associated with increased frequency of the G119S mutation in *An. gambiae* s.l. [23,24]. Still, there is not enough information on its geographical scale and intensity. 

The present study aims to characterize the resistance status of *Anopheles gambiae* s.l. to bendiocarb (carbamate) and malathion (organophosphate) in four epidemiological settings in Cameroon and to explore the molecular basis of resistance.

## 2. Results

### 2.1. Bioassays Results

A total of 960 *Anopheles* specimens from laboratory strains (Kisumu and Ngousso) and 2490 field specimens from Kékem (n = 160), Njombé (n = 200), Belabo (n = 200), Kaélé (n = 400), Tibati (n = 400), Bertoua (n = 570) and Santchou (n = 560) were exposed to determine their susceptibility profile against bendiocarb and malathion insecticide. The two laboratory strains were all susceptible to bendiocarb and malathion with mortality rate of 100%, demonstrating that the impregnated paper was good. The susceptibility profile was year- and insecticide-dependent in each locality (Figure 1). During the year 2021, in Santchou and Kekem, all tested mosquitoes were resistant to both insecticides (Figure 1). In Belabo, Bertoua and Tibati mosquitoes were found to be resistant to bendiocarb (mortality rate: 80%; 77.77%; 76% respectively) and fully susceptible to malathion (mortality rate: 98%; 99%; 100% respectively). Mosquitoes from Kaélé and Njombé were fully susceptible to both insecticides (mortality rate > 98%) (Figure 1).

In addition, the mortality rate of mosquitoes varied significantly with year, except in Santchou, where no significant difference was found regarding Bendiocab (χ^2^ = 3.558, *p* = 0.1687). Samples tested in 2020 in Bertoua and Tibati were found to be significantly more susceptible to bendiocarb (mortality rate 95% to 100%) compared to those collected in 2019 and 2021 (mortality rate varying from 31% to 78%).

### 2.2. Molecular Identification of Species of the Anopheles gambiae Complex

The genotyping of 637 specimens for species identification showed the presence of three members of the *An. gambiae* complex in the study sites, i.e., *An. coluzzii*, *An. gambiae* and *An. arabiensis*. In Tibati only *An. coluzzii* was recorded, whereas in Kaélé, *An. arabiensis*, *An. gambiae* and *An. coluzzii* were present with, however, a predominance of *An. coluzzii*. In Santchou and Kékem *An. gambiae* was the only species recorded, while in Bertoua, Njombé and Belabo both *An. gambiae* and *An. coluzzii* were recorded. *Anopheles gambiae* was the predominant species in Bertoua and Belabo, whereas *An. coluzzii* was slightly more abundant than *An. gambiae* in Njombé (Table 1).

### 2.3. Screening of Target Site Mutations (Ace-1 G119S)

The (G119S) mutation was detected in all sites. In Njombé the G119S mutation was detected mainly in bendiocarb and malathion survivors (75.5–100.0%) and at a very low frequency (4.5%) in the unexposed population. In Kékem bendiocarb survivors showed a frequency of 84.6% and unexposed mosquitoes a lower frequency (41.1%). The Bélabo population presented a frequency of 54.1% and 72.3% in bendiocarb and malathion survivors, respectively; it was also detected in unexposed mosquitoes in a frequency of 10.25%. In Tibati, a frequency of 81.3% was recorded in Bendiocarb survivors, whereas it was not detected in unexposed mosquitoes. In Bertoua, bendiocarb and malathion survivors showed frequencies of 71.0% and 77.1%, respectively; unexposed mosquitoes exhibited a frequency of 36.1%. In the Santchou population, the mutation was found in frequencies > 85.0% in bendiocarb and malathion survivors, as well as unexposed mosquitoes (Table 2).

### 2.4. Expression Analysis of Detoxification Genes

The expression profile of different detoxification genes known to confer resistance was analyzed. *Cyp6p3, Cyp6m2, Cyp9k1, Cyp6p4, Cyp6z1, Gste2* and *Cyp4g16* were among genes that were significantly overexpressed in at least one study population. More precisely, in the Njombé population (Figure 2A)*, Cyp9k1* and *Cyp6z1* were significantly overexpressed (>8.0 and >1.6 folds, respectively). In Kékem the *Cyp6m2* and *Cyp9k1* genes were both overexpressed by >3.0 folds (Figure 2B). The Kaélé population (Figure 2C) showed a marked overexpression of *Cyp6p4* (>10 folds) and *Gste2* (>5.0 folds). In Bélabo, *Cyp6p3* and *Cyp6p4* were overexpressed by >1.90 folds (Figure 2D). The Tibati population (Figure 2E) showed an overexpression of only *Cyp4g16* (>1.60 folds). In Bertoua, *Cyp9k1* and *Cyp6p4* were significantly overexpressed (>3.0 and >2.0 folds, respectively). The Santchou population showed a >1.6-fold overexpression of *Cyp6z1* (Figure 2G).

## 3. Discussion 

The study’s objective was to assess the distribution of carbamate and organophosphate resistance in *An. gambiae* s.l. populations from different epidemiological settings across Cameroon. High resistance to both carbamates and organophosphates was detected and was consistent with previous reports [23,24,25]. Resistance to bendiocarb was largely spread and was detected in five out of the seven sites surveyed, whereas resistance to malathion was detected in the sites of Kékem and Santchou situated in western Cameroon. These sites are semi-rural settings where vast lands are used for the cultivation of vegetables, maize, cocoa, plantains and tomatoes all year round with the intensive application of pesticides [26]. It is possible that high resistance to bendiocarb and malathion detected in these sites is driven by the frequent use of pesticides in agriculture rather than insecticides used in public health. In Tibati, Bertoua and Belabo, mosquitoes were found resistant to bendiocarb, while mosquitoes from Njombé and Kaélé were fully susceptible to bendiocarb and malathion. The heterogeneous pattern of carbamates and organophosphates resistance in *An. gambiae* s.l. populations across the country could derive from different selective pressure in each site and habitat segregation by the species of the *An. gambiae* complex [27,28]. Similar geographical patterns of carbamate or organophosphate resistance in members of the *An. gambiae* complex have been reported in other African countries [22]. *Anopheles arabiensis*, *An. gambiae* and *An. coluzzii* species were recorded in the study sites and their presence followed an eco-epidemiological/site-dependent pattern. More precisely, *An. coluzzii* was the predominant species in Kaélé, Tibati and Njombé, whereas *An. gambiae* was predominant in Santchou, Bertoua, Belabo and Kékem. *An. arabiensis* was detected in low frequency in Kaelé and Tibati. The high frequency of *An. coluzzii* in the Sahelian and Sudano-Sahelian region of the country (Kaélé and Tibati) is in line with previous studies which indicated that this species is outpacing *An. arabiensis* and is gradually becoming the predominant species in the Sudano-Sahelian savannah zone [29,30]. 

In an effort to molecularly profile the evident carbamate and organophosphate phenotypic resistance we measured the frequencies of the Ace-1 G119S mutation, as well as the expression levels of genes associated with metabolic and cuticle resistance.

The Ace-1 G119S mutation was detected at a very high frequency in almost all sites. Studies conducted so far in Africa suggested that the presence of this mutation is mainly associated with resistance to carbamates [17,18,22,31]. The Ace-1 G119S mutation was detected in both *An. gambiae* and *An. coluzzii* with, however, a higher frequency in *An. gambiae*. A similar distribution pattern has been reported in many countries across West Africa [17,22]. The present findings are different from previous studies [23,25], which reported the mutation only in *An. gambiae*. It is possible that the Ace-1 G119S mutation in *An. coluzzii* may have emerged during recent years through introgression or independently. Different frequencies of the G119S allele have been reported between *An. gambiae* and *An. coluzzii* in areas of sympatry. It has been suggested that the difference in this mutation frequency could result from discrepancies in the selection pressure mosquitoes experienced in aquatic habitats [22]. Despite the fact that the study did not characterize aquatic habitats, studies assessing breeding habitats influence in the cities of Yaoundé and Douala in Cameroon suggested that mosquitoes deriving from agricultural cultivated areas were more tolerant to different insecticides families, including pyrethroids and carbamates, compared to those originating from unpolluted sites [19,32]. 

In addition to the Ace-1 mutation the expression profiles of genes implicated in metabolic and cuticle resistance were assessed. Regarding carbamate resistance, *Cyp6p3* was significantly overexpressed in the Bélabo population, where phenotypic resistance to bendiocarb was also detected. This particular P450 gene has been previously shown to metabolize bendiocarb [33,34]. Regarding organophosphate resistance, in Santchou, a site with low mortality rates for malathion, the overexpression of *Cyp6p4* was detected. *Cyp6p4* has been previously shown to metabolize malathion, as well as other organophosphates (pyrimiphos-mehtyl, fenitrothion); however, a link to resistance cannot be safely concluded, and organophosphate metabolism by *Cyp6p4* could lead either to the activation of more toxic metabolites or detoxification, and the relative activation versus detoxification contribution is not clear. This is also the case for *Cyp6p3* and *Cyp6m2* with data for the latter pointing towards activation rather than detoxification. The glutathione S-transferase gene *gste2* was found to be markedly overexpressed in the Kaélé population. Since, *gste2* has been strongly linked with resistance to the organophosphate fenitrothion [34] this constitutes an alarming factor. Our expression analysis revealed also the significant overexpression of genes that are known pyrethroid (*Cyp6m2*, *Cyp9k1*, *Cyp6p4*) and organochlorine (*gste2, Cyp6z1, Cyp6m2*) metabolizers [9]. Resistance to these insecticides is well documented in Cameroon [35]. All the above show that cross-resistance and multi-resistance is in place in the study’s populations. Regarding cuticle resistance, *Cyp4g16*, a functional oxidative decarboxylase gene known to catalyze epicuticular lipid biosynthesis and contribute to insecticide resistance via the enrichment of the CHC content, thus reducing insecticide uptake was also found to be upregulated in one population (Tibati) [36].

## 4. Methods

### 4.1. Study Site

Mosquitoes were collected from seven localities in Cameroon: Kékem, Njombé, Belabo, Kaélé, Tibati, Bertoua and Santchou (Figure 3). A detailed characteristic of each site is provided in Table 3. Physical characteristics of sites with superscript (^a^) is published in Kala-Chouakeu et al. [37].

### 4.2. Collection of Mosquito Larvae, Rearing, and Conservation

The field sampling of anopheline larvae and processing was conducted during the long rainy season in all the sites. It was carried out once yearly in August 2019, August 2020 and November 2021 in Bertoua and Santchou, whereas in the remaining sites the sampling and processing of mosquitoes was performed once or twice (Kékem, Njombé, Belabo, Kaélé, Tibati) because of the unavailability of funds (Figure 3). Larval collections were undertaken in different habitats including temporary water collections, puddles, and semi-permanent sites. In each study site, collected samples from different breeding habitats were pooled per site and reared into adults in the insectary created in the field. Larvae were fed with TetraMin® fish food until pupae. Pupae were collected in a cardboard cup and placed in netting cages for adult emergence. After emergence, adults were offered sugar solution until processing. A subset of 30–40 unexposed, non-blood fed, 3–5-day-old female *An. gambiae* s.l. from different sites and susceptible strains (the *An. gambiae* Kisumu strain and the *An. coluzzii* Ngousso strain) were preserved in RNAlater™ for the characterization of the molecular mechanisms of insecticide resistance. The same sample was also used for species identification. Another subset of about 120–150 mosquitoes per population was used for insecticide bioassays; survivors after exposure to insecticide were preserved in 70% alcohol and used for the confirmation of molecular species identification and detection of G119S.

### 4.3. Insecticide Bioassay

Adult female *An. gambiae* s.l. reared from larval collections in different sites were tested alongside the susceptible laboratory strains Kisumu and Ngousso against two insecticides (bendiocarb 0.01% and malathion 5%) following WHO guidelines [38]. *An. gambiae* s.l. females aged 3–5 days reared from larvae collected on the field were placed in batches of 20 to 25 mosquitoes per tube and left for observation for one hour. After this period, mosquitoes were transferred to four different tubes with insecticide-impregnated papers and exposed for 1 h. The susceptible laboratory strains (*An. gambiae* (Kisumu strain) and *An. coluzzii* (Ngousso strain)) were used as control to assess the quality of the impregnated papers. The number of mosquitoes knocked down by the insecticide was recorded after 1h of exposure; then, mosquitoes were fed with a 10% sugar solution and the number of dead mosquitoes was recorded 24 h post-exposure. Mosquitoes were considered resistant when the mortality rate was <90% and susceptible when the mortality rate was ≥98%, and resistance status needed further scrutiny when the mortality rate was <98% and >90% [39].

### 4.4. Mosquito Processing

#### 4.4.1. Total RNA and DNA Extraction from Mosquito Pools

Using a magnetic beads-based approach with the MagSi kit (MagnaMedics Diagnostics B.V., Geleen, Netherlands), total genomic material (RNA and DNA) was extracted from 687 mosquitoes (N = 90 from Kékem, N = 64 from Njombé, N = 94 from Belabo, N = 70 from Kaélé, N = 90 from Tibati, N = 130 from Santchou and N = 89 from Bertoua, N = 30 from Kisumu and N = 30 from Ngousso, susceptible strains). Mosquitoes were pooled for extractions (up to N = 10 mosquitoes per pool). The quantity and purity of DNA and total RNA were assessed spectrophotometrically via Nanodrop measurements. The average NA concentration was 58.96 ± 5.4 ng/μL and the average A_260_/A_280_ ratio, indicative of sample purity, was 2.3 ± 0.05. The quality of RNA was assessed by 1.0% *w*/*v* agarose gel electrophoresis. Representative samples are shown in Appendix A.

#### 4.4.2. Species Identification 

Mosquitoes used for species identification included those exposed to insecticide (survivors, dead) and some not exposed to insecticides. Mosquitoes from different years were analyzed. Species identification at the molecular level was performed using the TaqMan assays described in the Innovative Vector Control Consortium (IVCC) Vector Population Monitoring Tool (VPMT) [39] with modifications described in Wipf et al. [40] for the detection of *An. coluzzii* and *An. gambiae* species (Primers: S200_X6.1_F: TCGCCTTAGACCTTGCGTTA, S200_X6.1_R: CGCTTCAAGAATTCGAGATAC, Probes:Pcoluzzii: HEX-ACCGCGCCGCCATACGTAGGA-BHQ1 and Pgambiae: FAM-ATGTCTAATAGTCTCAATAGT -MGB).

### 4.5. Genotyping of G119S Mutation 

The G119S mutation was analyzed using the TaqMan assay (Primers: F: GGCCGTCATGCTGTGGAT and R: GCGGTGCCGGAGTAGA; Probes: Pwt: HEX-TTCGGCGGCGGCT-MGB and Pmut: FAM-TTCGGCGGCAGCT-MGB), as described in VPMT [39]. The percentage of allele frequency for the previously mentioned traits in mosquito pools was calculated with regression models using the protocol described in Mavridis et al., 2018 [41]. Briefly, a standard curve for G119S using plasmid sequences with known % frequencies was constructed. It was then used for the calculation of % G119S frequencies for the unknown populations by transforming the dCt (CtF_AM_-Ct_HEX_ values) to allelic frequencies via the equation of the standard curve’s regression model (Appendix A).

### 4.6. Resistance Gene Expression Analyses

The reverse transcription quantitative PCR (qRT-PCR) 3-plex TaqMan® assays described in Mavridis et al., 2019 [42] were used for the quantification of seven detoxification genes (*Cyp6p3*, *Cyp6m2*, *Cyp9k1*, *Cyp6p4*, *Cyp6z1*, *Gste2*, *Cyp6p1*) that have been strongly associated with metabolic resistance and one oxidative decarboxylase (*Cyp4g16*) that is implicated in cuticle resistance [42]. The list of primers and probes that were used are listed in Appendix A. Reactions were performed in the Viia7 Real-Time PCR system (Applied Biosystems) using the following thermal cycle parameters: 50 °C for 15 min, 95 °C for 3 min and 40 cycles of 95 °C for 3 s and 60 °C for 30 s. The QuantStudio^TM^ Real-Time PCR system v1.3 (Applied Biosystems) software was used for the calculation of Ct values for each reaction.

### 4.7. Statistical Analysis

Using the method developed by Pfaff [43], fold-changes, 95% CIs and statistical significance were calculated, while graphs were constructed with the SigmaPlot software (v12.0). The mortality rate was expressed as the ratio between the number of mosquitoes that were found dead or not capable of standing on their legs and exposed ones. Confidence intervals were computed using Medcalc. The comparison of mortality rate and fold change was performed using chi square and Student’s t-test, respectively.

## 5. Conclusions

This study indicates the heterogeneous distribution of carbamate and organophosphate resistance in *An. gambiae* s.l. populations across Cameroon. Both Ace-1 G119S mutation and the overexpression of genes associated with carbamate/organophosphate resistance were detected. The current expansion of carbamates and organophosphate resistance could seriously compromise future malaria control measures based on the use of organophosphates and carbamates in IRS. Addressing gaps affecting malaria control by improving surveillance activities on the vector is becoming crucial to ensure the success of malaria elimination efforts in Cameroon. 

## Figures and Tables

**Figure 1 pathogens-11-00824-f001:**
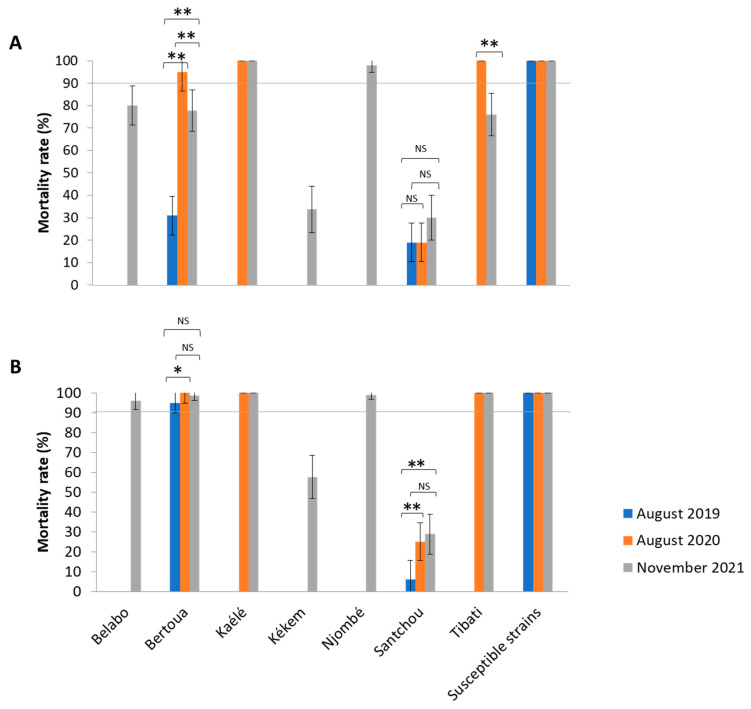
Mortality rate (%) of field collected and laboratory strains of *Anopheles gambiae* s.l. to bendiocarb 0.01% (carbamate) (**A**) and malathion 5% (organophosphate) (**B**) over time. At least 20–25 mosquitoes in four replicates were exposed per insecticides per site and during each collection period. Color represents the collection period (blue: August 2019; orange: August 2020 and grey: November 2021). The line (90%) represents the threshold of susceptibility (resistant when the mortality rate was <90% and susceptible when the mortality rate was >90%. The upper script (*) *p* < 0.05, (**) *p* < 0.01; (NS) non-significant. The bars represent standard error.

**Figure 2 pathogens-11-00824-f002:**
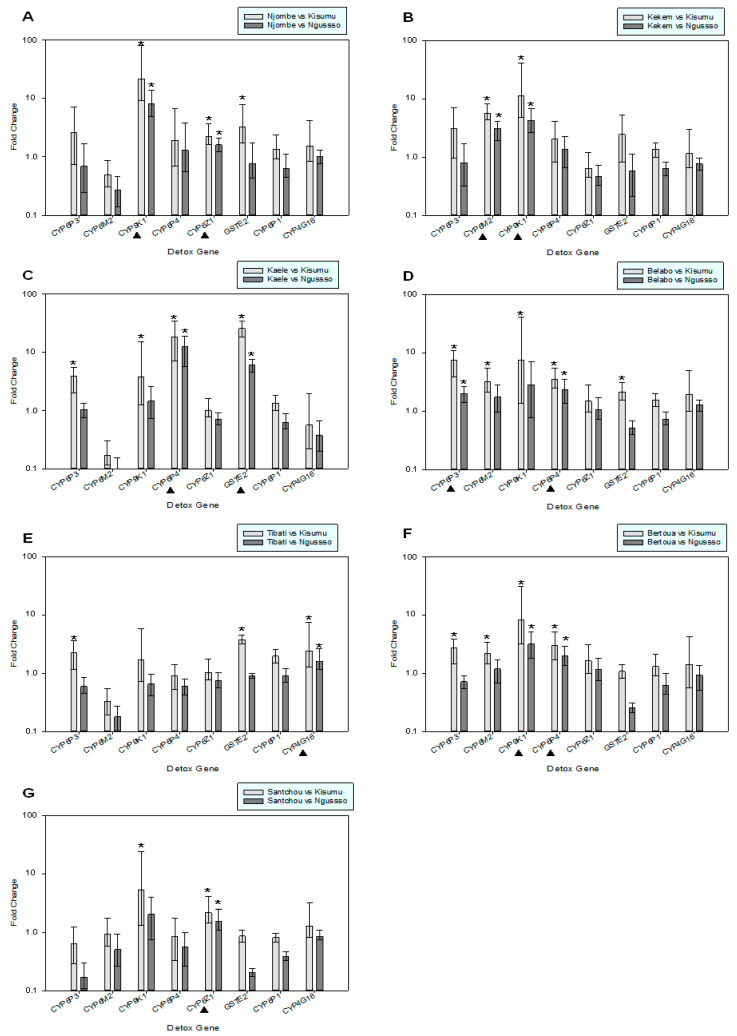
Expression analysis of genes associated with insecticide resistance in the study’s populations (parts **A**–**G**). Error bars indicate 95% CIs. * denotes genes that showed statistically significant upregulation. ▲ denotes genes that showed consistent upregulation compared to both susceptible strains.

**Figure 3 pathogens-11-00824-f003:**
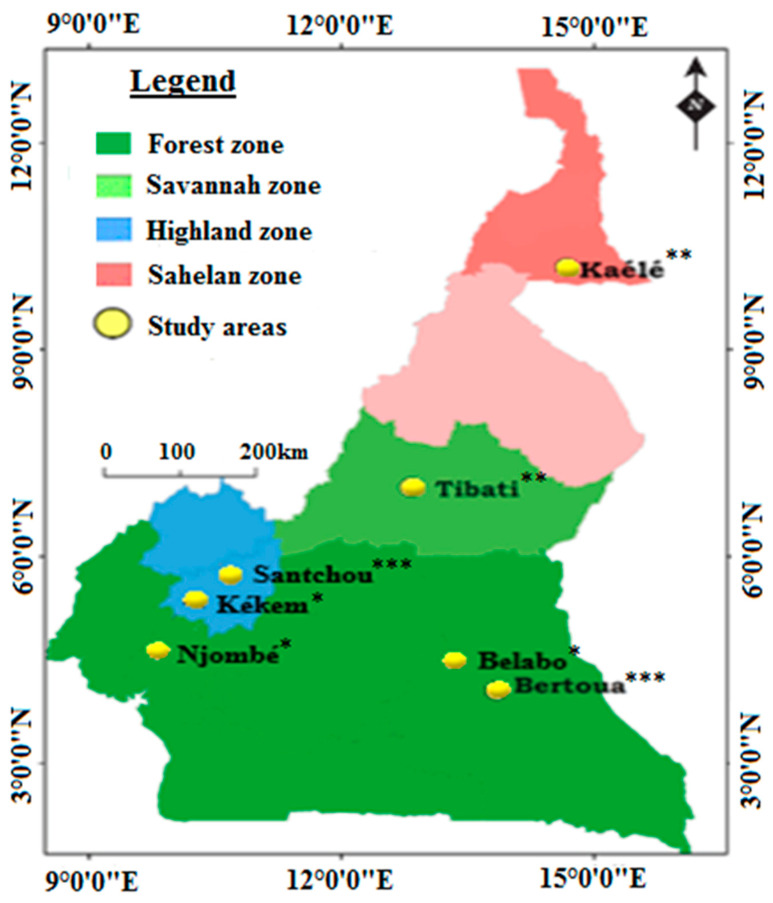
Map of Cameroon showing the study sites. Number of collection periods (* = once, ** = twice, *** = thrice).

**Table 1 pathogens-11-00824-t001:** Distribution of *An. gambiae* s.l. species at the different study sites.

Population/Sites	Sites Characteristics	Sample Size	Species ID	% Frequency
Kisumu	Susceptible strain	40	*An. gambiae*	100
Ngousso		40	*An. coluzzii*	100
Tibati	Sahelo-Sudanese (humid savannah)	80	*An. coluzzii*	100
Kaélé	Sahelian zone		*An. arabiensis*	10.0
		60	*An. gambiae*	4.0
			*An. coluzzii*	86.0
Kékem	Highland grassfields	80	*An. gambiae*	100
Santchou		120	*An. gambiae*	100
Bertoua	Forest zone	79	*An. gambiae*	87.0
			*An. coluzzii*	13.0
Njombé		54	*An. gambiae*	40.9
			*An. coluzzii*	59.1
Belabo		84	*An. gambiae*	82.0
			*An. coluzzii*	18.0

**Table 2 pathogens-11-00824-t002:** Ace-1 G119S mutation allelic frequencies in the different samples (mean ± SE).

PopulatioN	Category	Sample Size	% G119S (ace-1)
Kisumu	Susceptible lab strain	40	0.0 ± 0.0
Ngousso	Susceptible lab strain	40	0.0 ± 0.0
Njombé	Bendiocarb survivors	3	75.5
	Malathion survivors	1	100.0
	Unexposed	20	4.5 ± 2.5
Kékem	Bendiocarb survivors	20	84.6 ± 5.1
	Unexposed	20	41.1 ± 18.3
Bélabo	Bendiocarb survivors	20	54.1 ± 9.6
	Malathion survivors	4	72.3
	Unexposed	20	10.25 ± 5.9
Tibati	Bendiocarb survivors	20	81.3 ± 18.7
	Unexposed	20	0.0 ± 0.0
Bertoua	Bendiocarb survivors	18	71.0 ± 9.9
	Malathion survivors	1	77.1
	Unexposed	20	36.1 ± 9.4
Santchou	Bendiocarb survivors	20	87.3 ± 1.75
	Malathion survivors	40	86.93 ± 8.0
	Unexposed	20	87.9 ± 12.2

**Table 3 pathogens-11-00824-t003:** Physical characteristics of the study sites.

Study sites	Kékem	Njombé	Belabo	Kaélé ^a^	Tibati ^a^	Santchou ^a^	Bertoua ^a^
Administrative region	West	Littoral	East	Far North	Adamawa	West	East
Coordinates	5°10′ N, 10°02′ E	4°64′ N, 9°67′ E	4°56′ N, 13°18′ E	10°50′ N, 14°56′ E	12°37′ N, 12°37′ E	5°58′ N, 9°58′ E	4°34′ N, 13°41′ E
Domain	Highland Grassfields	Forest zone	Forest zone	Sahelian zone	Sahelo soudanese	Highland Grassfields	Forest zone
Climate	Equatorial	Equatorial	Subtropical	Sahelian	Tropical humid	Equatorial	Subtropical
Seasons	Dry season (November to March), rainy season (April to October)	Dry season (December to February), rainy season (March to November)	Dry season (December to March and July), rainy season (March to June and August to November)	Dry season (October to May), rainy season (June to September)	Dry season (November to April), rainy season (May to October)	Dry season (November to March), rainy season (April to October),	Dry season (December to March and July), rainy season (March to June and August to November)
Vegetation	Grassland	Grassland	Evergreen degraded forest	Wooded Savanah	Grassy Savanah	Grassland	Evergreen degraded forest

^a^ Physical characteristics of sites published in Kala-Chouakeu et al. [37].

## Data Availability

All data have been made available in the manuscript and Appendix A.

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
