# Peer review of "Bendiocarb and Malathion Resistance in Two Major Malaria Vector Populations in Cameroon Is Associated with High Frequency of the G119S Mutation (Ace-1) and Overexpression of Detoxification Genes"

_pathogens, 2022, doi:10.3390/pathogens11080824_

Round 1

Reviewer 1 Report

The control of vector-borne diseases, particularly malaria, relies mainly on the use of insecticide-based tools. Therefore, it is essential to assess the susceptibility of vectors to the chemicals used in order to design efficient control strategies. The study by Ngangue-Siewe et al. has the merit of studying the susceptibility of anopheles to carbamates and organophosphates in order to assess the possibility of using them in addition to pyrethroids, which are mainly used for IRS and in bed nets.

Their work certainly deserves to be published, but some clarifications should be made regarding the methodology and the presentation of the results

Major remarks:

In the sample collection section, on two sites, the collections were made over three years and during the rainy season. On the other sites of collection, no indication was given on the collection period (rainy or dry season) and especially why the same design was not applied on all sites to have uniform collections during the whole study period. Is there a seasonal dynamic of the different vectors?  If so, could the composition of the vectors as well as the susceptibility profile to bendiocarb and malathion evolve according to the season?

In the bioassay section, the process of susceptibility testing is not accurately reported. There is no information in this section on what basis a population is considered susceptible or resistant to insecticides.

The exposure of mosquitoes to the paper in the tubes is not well defined, what is the concentration of insecticide on each paper? Were 4 tubes (4 papers) used for each insecticide?

In the DNA and RNA extraction section, although the number of samples is indicated, there is no indication of where they came from (study site and number per site), what part of the mosquito was used to extract the genomic material. In addition, the authors indicate that they have verified the quality of both RNA and DNA, but no indication is provided regarding these tests results. It is also the same for sections 5 and 6 of the methodology where the authors systematically refer to other publications to explain their methodology. They should give more information on the course of their methodology.

In the results section, the presentation of data of the bioassays should be improved by indicating in the text the mortality rate obtained with each insecticide for each population. This would be more indicative than just talking about susceptible or resistant population, especially since the authors did not indicate the threshold of susceptibility to insecticides in the methodology. The understanding of the paragraph between lines 105 and 109 is not clear. Indeed, in figure 1 we can see that for bendiocarb there is a variation in mortality during the last year compared to the two previous ones? Do the authors mean that the variation is not significant? In addition, it would be nice to put on figure 1 the significant variations of mortality and possibly the threshold of susceptibility.

Concerning table 1 indicating the different species of anopheles encountered, was this done from exposed mosquitoes or from mosquitoes kept after their emergence? How this sampling was made? Does it take into account the densities of anopheles at each site and the years of collection for sites where samples were taken over 2 or 3 years? The same goes for table 3.

The same thing is valid for the analysis of the expression of detoxification genes, how does it evolve from one year to the next at sites with 2 to 3 sampling.

Also did the authors tried to see if there is a difference in the expression of metabolic genes between the two susceptible strains? Because we can see that, despite the fact that these two strains are susceptible to insecticides, the expression of their metabolic genes is different.

Minor remarks

Are the susceptible strains described in line 245 identical to the one described in line 258?  If so, indicate the identity of these strains from line 245 onwards

What is the purpose of the observation step described in line 256?

Author Response

Organisation de Coordination pour la

lutte contre les Endémies en Afrique Centrale

Malaria Research Laboratory

Dr ANTONIO-NKONDJIO Christophe

Email: Antonio_nk@yahoo.fr

To

The Editor in chief

Pathogens

Point-by-point response to reviewers

Dear Sir/Madam,

I would like to thank the reviewers for their comments and suggestions to improve the quality of the manuscript. All requested changes were undertaken accordingly.

Comments and Suggestions for Authors

Major remarks:

Comment:  In the sample collection section, on two sites, the collections were made over three years and during the rainy season. On the other sites of collection, no indication was given on the collection period (rainy or dry season) and especially why the same design was not applied on all sites to have uniform collections during the whole study period.

Response: We thank the reviewer for giving us the opportunity to clarify these issues. The collection was made during the rainy season in all sites.  Collections and sample processing were not done in all site during the three years because of the unavailability of funds.

Comment:  Is there a seasonal dynamic of the different vectors?  If so, could the composition of the vectors as well as the susceptibility profile to bendiocarb and malathion evolve according to the season?

Response: We thank the reviewer for this remark. There is not much seasonal variation in species composition in the different sites.  

Comment:  In the bioassay section, the process of susceptibility testing is not accurately reported. There is no information in this section on what basis a population is considered susceptible or resistant to insecticides.

Response: We thank the reviewer for this observation. The information was added at the end of section on “insecticide bioassay “Mosquitoes were considered resistant when the mortality rate was < 90% and susceptible when the mortality rate was ≥ 98% and resistance status need further checking when mortality rate < 98% and ˃ 90%”.

Comment: The exposure of mosquitoes to the paper in the tubes is not well defined, what is the concentration of insecticide on each paper? Were 4 tubes (4 papers) used for each insecticide?

Response: We thank the reviewer for this remark. The concentration of Bendiocarb is 0.01% and the one of Malathion is 5%. The information was added in the section on “Insecticide bioassay “

Comment: In the DNA and RNA extraction section, although the number of samples is indicated, there is no indication of where they came from (study site and number per site), what part of the mosquito was used to extract the genomic material. In addition, the authors indicate that they have verified the quality of both RNA and DNA, but no indication is provided regarding these tests results. It is also the same for sections 5 and 6 of the methodology where the authors systematically refer to other publications to explain their methodology. They should give more information on the course of their methodology.

Response: We thank the reviewer for giving us the opportunity to clarify these issues. Following the reviewer’s suggestions, we have now added the following information in our revised manuscript, marked with red color here:

-4.4.1 Total RNA and DNA extraction from mosquito pools

Using a magnetic beads-based approach with the MagSi kit (MagnaMedics Diagnostics B.V.), total genomic material (RNA and DNA) was extracted from 687 mosquitoes (N=90 from Kékem, N=64 from Njombé , N=94 from Belabo, N=70 from Kaélé, N=90 from Tibati, N=130 from Santchou and N=89 from Bertoua, N=30 from Kisumu and N=30 from Ngousso susceptible strains). Mosquitoes were pooled for extractions (up to N=10 mosquitoes per pool). The quantity and purity of DNA and total RNA were assessed spectrophotometrically via Nanodrop measurements. The average NA concentration was 58.96 ± 5.4 ng/μL and the average A260/A280 ratio, indicative of sample purity, was 2.3 ± 0.05. The quality of RNA was assessed by 1.0% w/v agarose gel electrophoresis. Representative samples are shown in Supplementary Figure S1.

Suppl figure S1 Agarose gel (1.0% w/v) electrophoresis for total RNA of representative samples (N=6). The presence of distinct ribosomal bands (28S, 18S, 5S from top to bottom) and the absence of degradation products shows that total RNA was intact, suitable for downstream analyses. 

-4.4.2. Species identification

Species identification at the molecular level was performed using the TaqMan assays described in the Innovative Vector Control Consortium (IVCC) Vector Population Monitoring Tool (VPMT) [40] with modifications described in Wipf et al [41] for the detection of An. coluzzii and An. gambiae species (Primers: S200_X6.1_F: TCGCCTTAGACCTTGCGTTA, S200_X6.1_R: CGCTTCAAGAATTCGAGATAC, Probes: Pcoluzzii: HEX-ACCGCGCCGCCATACGTAGGA -BHQ1 and Pgambiae: FAM-TGTCTAATAGTCTCAATAGT -MGB).

-5. Genotyping of G119S mutation

The G119S mutation was analyzed using the TaqMan assay (Primers: F: GGCCGTCATGCTGTGGAT and R: GCGGTGCCGGAGTAGA and Probes: Pwt: HEX- TTCGGCGGCGGCT-MGB and Pmut: FAM- TTCGGCGGCAGCT-MGB) described in VPMT [40].  The percentage of allele frequency for the previously mentioned traits in mosquito pools was calculated with regression models using the protocol described in Mavridis et al., 2018 [42]. Briefly, a standard curve for G119S using plasmid sequences with known % frequencies was constructed. It was then used for the calculation of % G119S frequencies for the unknown populations by transforming the dCt (CtFAM-CtHEX values) to allelic frequencies via the equation of the standard curve’s regression model (Suppl Figure S2)

Supplementary Figure S2 Standard curve for G119S using plasmid sequences with known % frequencies (A) and calculation of % G119S for an unknown population of the study as an example (B).

- 6. Resistance gene expression analyses

The reverse transcription -quantitative PCR (qRT-PCR) 3-plex TaqMan® assays described in Mavridis et al, 2019 [43] were used for the quantification of seven detoxification genes (Cyp6p3, Cyp6m2, Cyp9k1, Cyp6p4, Cyp6z1, Gste2, Cyp6p1) that have been strongly associated with metabolic resistance and one oxidative decarboxylase (Cyp4g16) that is implicated in cuticle resistance [43]. The list of primers and probes that were used are listed in Suppl. Table S1. Reactions were performed in the Viia7 Real-Time PCR system (Applied Biosystems) using the following thermal cycle parameters: 50 °C for 15 min, 95 °C for 3 min, and 40 cycles of 95 °C for 3 sec and 60 °C for 30 sec. The QuantStudioTM Real-Time PCR system v1.3 (Applied Biosystems) software was used for the calculation of Ct values for each reaction.

Comment: In the results section, the presentation of data of the bioassays should be improved by indicating in the text the mortality rate obtained with each insecticide for each population. This would be more indicative than just talking about susceptible or resistant population, especially since the authors did not indicate the threshold of susceptibility to insecticides in the methodology.

Response: we thank the reviewer for pointing this out. we have revised the manuscript accordingly

Comment: The understanding of the paragraph between lines 105 and 109 is not clear. Indeed, in figure 1 we can see that for bendiocarb there is a variation in mortality during the last year compared to the two previous ones? Do the authors mean that the variation is not significant? In addition, it would be nice to put on figure 1 the significant variations of mortality and possibly the threshold of susceptibility.

Response: we thank the reviewer for this observation. Information added in figure1.

Comment:  Concerning table 1 indicating the different species of anopheles encountered, was this done from exposed mosquitoes or from mosquitoes kept after their emergence? How this sampling was made? Does it take into account the densities of anopheles at each site and the years of collection for sites where samples were taken over 2 or 3 years? The same goes for table 3.

Response:  we thank the reviewer for this question. The following information was added in “species identification” in the method section “Mosquitoes used for species identification included those exposed to insecticide (survivors, dead) and some not exposed to insecticides. Mosquitoes from different years were analysed.”

Comment:  The same thing is valid for the analysis of the expression of detoxification genes, how does it evolve from one year to the next at sites with 2 to 3 sampling.

Response: We thank the reviewer for this observation. We agree that it would be interesting to see how this mechanism evolves over different time points and we will include this in our future studies. In the current study however, there was only one sample (the same year) analysed per location.

Comment: Also did the authors tried to see if there is a difference in the expression of metabolic genes between the two susceptible strains? Because we can see that, despite the fact that these two strains are susceptible to insecticides, the expression of their metabolic genes is different.

Response: We thank the reviewer for pointing this out. Indeed, we are aware that there are differences in the expression profiles of susceptible strains, as it can been retrieved from several publications that use more than one susceptible strain for comparison. This is the reason we chose to include both susceptible strains for comparison with resistant strains, i.e., to increase the strictness for detecting metabolic gene upregulation. In other words, only if a population shows statistically significant upregulation compared to both susceptible strains this is considered biologically relevant and commented upon. However, comparison between susceptible strains was beyond our study’s goals.

Minor remarks

Comment: Are the susceptible strains described in line 245 identical to the one described in line 258?  If so, indicate the identity of these strains from line 245 onwards.

Response: We thank the reviewer for this remark, yes these are the same susceptible strains, we have revised accordingly. 

Comment: What is the purpose of the observation step described in line 256?

Response: we thank the reviewer for this question.  The purpose of the observation step described in this line is to discard from the test mosquitoes those who are moribund or not flying.

All requested changes were done accordingly and are indicated by tracked changes in the main text. Thanks

Reviewer 2 Report

Overall, the manuscript is written well and is easy-to-understand. I only have the following minor suggestions. 

Abstract

Please italicize Anopheles gambiae (and An. coluzzii) in the abstract and throughout the manuscript.

Introduction

Line 56: please change ‘and humans’ to ‘including humans’

Line 65: please change ‘achieve’ to ‘achieved’

Line 66: please change ‘alternative’ to ‘alternatives’

Materials and Methods

Line 226: please change ‘in seven localities’ to ‘from seven localities’

Results

No comments.

Discussion

Line 169: please change ‘where’ to ‘were’

Line 183: ‘molecularly’

Line 184: ‘phenotypic’ and ‘measured’

Author Response

Organisation de Coordination pour la

lutte contre les Endémies en Afrique Centrale

Malaria Research Laboratory

Dr ANTONIO-NKONDJIO Christophe

Email: Antonio_nk@yahoo.fr

To

The Editor in chief

Pathogens

Point-by-point response to reviewers

Dear Sir/Madam,

I would like to thank the reviewers for their comments and suggestions to improve the quality of the manuscript. All requested changes were undertaken accordingly.

Abstract

Please italicize Anopheles gambiae (and An. coluzzii) in the abstract and throughout the manuscript.

Response: we thank the reviewer for the remarks. Changes done

Introduction

Line 56: please change ‘and humans’ to ‘including humans’

Line 65: please change ‘achieve’ to ‘achieved’

Line 66: please change ‘alternative’ to ‘alternatives’

Response: Changes done

Materials and Methods

Line 226: please change ‘in seven localities’ to ‘from seven localities’

Response: done

Results

No comments.

Discussion

Line 169: please change ‘where’ to ‘were’

Line 183: ‘molecularly’

Line 184: ‘phenotypic’ and ‘measured’

Response: done

All requested changes were done accordingly and are indicated by tracked changes in the main text. Thanks

Round 2

Reviewer 1 Report

The authors have clarified the issues raised in the first draft of the manuscript. These corrections have significantly improved the quality of the manuscript which deserves to be published.